# Lack of head sparing following third-trimester caloric restriction among Tanzanian Maasai

Christopher D. Powell[1], Warren M. Wilson[2], Godwin Olesaningo[3], Mange Manyama[4,5], Heather Jamniczky[2,6], Richard Spritz[7], James C. Cross[8,9], Kenneth Lukowiak[10], Benedikt Hallgrimsson[1], Paula N. Gonzalez[11]*

1 Department of Cell Biology & Anatomy, Cumming School of Medicine, University of Calgary, Calgary, Alberta, Canada, 2 Department of Anthropology and Archaeology, University of Calgary, Calgary, Alberta, Canada, 3 Endulen Hospital, Ngorongoro Conservation Area, Tanzania, 4 Catholic University of Health and Allied Sciences, Mwanza, Tanzania, 5 Division of Medical Education, Weill Cornell Medicine, Qatar, 6 McCaig Bone and Joint Institute, University of Calgary, Calgary, Alberta, Canada, 7 Department of Pediatrics and Human Medical Genetics and Genomics Program, University of Colorado School of Medicine, Denver, Colorado, 8 Department of Comparative Biology and Experimental Medicine, Faculty of Veterinary Medicine, and the Alberta Children's Hospital Research Institute, University of Calgary, Calgary, Alberta, Canada, 9 Department Biochemistry & Molecular Biology, University of Calgary, Calgary, Alberta, Canada, 10 Hotchkiss Brain Institute, Cumming School of Medicine, University of Calgary, Canada, 11 Unidad Ejecutora de Estudios en Neurociencias y Sistemas Complejos (CONICET-Hospital El Cruce Dr. Nestor Kirchner-Uiversidad Nacional Arturo Jauretche), Buenos Aires, Argentina

* pgonzalez@fcnym.unlp.edu.ar

**Data Availability Statement:** Data are available from Dryad: https://doi.org/10.5061/dryad.18931zcv7.

## Abstract

The reduction of food intake during pregnancy is part of many cultural and religious traditions around the world. The impact of such practices on fetal growth and development are poorly understood. Here, we examined the patterns of diet intake among Maasai pregnant women and assessed their effect on newborn morphometrics. We recruited 141 mother-infant pairs from Ngorongoro Conservation Area (NCA) in Northern Tanzania and quantified dietary intake and changes in maternal diet during pregnancy. We obtained measurements of body weight (BW) and head circumference (HC) at birth. We found that Maasai women significantly reduced their dietary intake during the third trimester, going from an average of 1601 kcal/day during the first two trimesters to 799 kcal/day in the final trimester. The greatest proportion of nutrient reduction was in carbohydrates. Overall, 40% of HC Z-scores of the NCA sample were more than 2 standard deviations below the WHO standard. Nearly a third of neonates classify as low birth weight (< 2500g). HC was smaller relative to BW in this cohort than predicted using the WHO standard. This contrasts markedly to a Tanzanian birth cohort obtained at the same time in an urban context in which only 12% of infants exhibited low weight, only two individuals had HC Z-scores < 2 and HC's relative to birth weight were larger than predicted using the WHO standards. The surprising lack of head sparing in the NCA cohort suggests that the impact of third trimester malnutrition bears further investigation in both animal models and human populations, especially as low HC is negatively associated with long term health outcomes.

**Funding:** This project was supported by: National Institute of Dental and Craniofacial Research to RS and BH (Grant number: 1U01DE020054) https://www.nidcr.nih.gov/ Natural Sciences of Engineering Research Council of Canada to BH (Grant number: 238992-12) http://www.nserc-crsng.gc.ca/index_eng.asp No funding bodies had any role in study design, data collection and analysis, decision to publish, or preparation of the manuscript.

**Competing interests:** The authors have declared that no competing interests exist.

**Abbreviations:** FFQ, Food Frequency Questionnaire; NCA, Ngorongoro Conservation Area; TBA, Traditional Birth Attendant; HC, Head Circumference; BW, Body Weight.

## Introduction

Maternal nutrition has profound effects on fetal growth and development [1–5]. The World Health Organization (WHO) recommends 200–300 additional calories per day during second and third trimester to promote prenatal tissue synthesis, fat deposition, and general maintenance [6,7]. Undernourished women are at greater risk of developing placental insufficiencies and giving birth to growth-restricted infants, who in turn face greater risk of infant mortality and morbidity [8,9]. In many communities around the world women reduce food intake during pregnancy because of cultural and religious practices [10–18]. The consequences of such practices on neonatal outcomes are of great interest to biomedically based interventions that seek to improve maternal and fetal health.

Here, we focus on the cultural practice of third trimester dietary restriction followed by Maasai (a Nilotic ethnic group settled in Kenya and Tanzania) women during pregnancy [19,20]. The origins of this practice of Maasai ethno-medicine are unknown, but appear to be motivated by a desire to avoid obstetric complications by limiting neonate size. Such beliefs occur in a variety of cultural settings around the world [21]. The "obstetric dilemma", is the hypothesized evolutionary tradeoff between human female pelvic dimensions and infant head size due to opposing selection for larger brains at birth and the biomechanical costs of a larger birth canal selection [22–24]. Whether conscious or not, the cultural practice of maternal dietary restriction may target the obstetric dilemma by reducing neonatal weight and head size in order to reduce the risk of maternal and neonatal mortality at birth. This is supported by self-reported beliefs that pregnant women must decrease their food intake to prevent large babies, which, in turn, eases labour and delivery [19,25,26]. Among the Maasai, pregnant women are also encouraged to avoid foods such as milk or fatty meats [27] and to ingest traditional medicines with emetic properties (A*lbizia anthelmintica*, *Warbugia ugandensis*) [19]. The impact of these practices on maternal nutritional status is aggravated by the fact that women maintain high metabolic demands by continuing household domestic workload, such as collecting water and firewood for consumption during the postnatal period [20]. The impact of these practices on neonatal outcomes have not been assessed and are poorly understood.

Growth restriction often affects the brain to a lesser extent than body mass [28,29]. This phenomenon, termed brain-sparing, is a common feature of growth-restricted neonates in both humans and in rodent models, and it is thought to reflect selection to mitigate the high functional cost of reduced brain growth and compromised neural development [30]. However, several experimental and population studies show that early exposure to calorie and protein restriction, even when it is moderate, affects the cellular composition, brain connectivity, and growth of brain components, which can persist long after conditions are improved [31,32]. Fetal growth restriction may not only be associated with increased risk of morbidity and mortality, but also with undernutrition-related cognitive impairment and neurological disorders [33–35].

The timing of nutritional stress during pregnancy may be an important determinant of birth outcomes but little is known about the importance of this variable in humans. Here, we quantify caloric and macronutrient intake of Maasai women across pregnancy and assess the effects on the growth status of their newborns by measuring body weight (BW) and head circumference (HC). We hypothesize that third trimester food restriction is associated with reduction in birth weight and head circumference at birth. HC is used here as a proxy of total brain volume since both variables are highly correlated during early life [36,37].

## Methods

### Study population

The present study focuses on Maasai pastoralists inhabiting the Ngorongoro Conservation Area (NCA) (S1 Fig). Maasai are among 60,000 full-time NCA residents who subsist on live-stock husbandry and intermittent cultivation [38–40]. Access to natural resources is limited by climate, exclusion from water sources, human population growth, and disease [41]. In addition, federal policy limits opportunities for small-scale cultivation available to NCA Maasai communities [40].

In the NCA area the climate is bimodal, with seasonal precipitations in November and December, followed by longer periods of rainfall from early March to late May [42,43]. During the dry season, acute water shortages require women to travel up to 10 km daily to gather approximately 10–20 L of water for domestic use (personal communication and field observations, 2008–2010). During this period, it is common for significant numbers of cattle to succumb to starvation, leading to diminished milk production, a traditional staple of Maasai diet [43,44]. Although seasonal food insecurity is initially relieved during the rainy seasons, damp conditions and lower temperatures are associated with respiratory infections and zoonotic diseases that further reduce cattle populations [45,46].

While pastoralism remains the dominant mode of subsistence, NCA Maasai communities organize and transact through a closed market system of commodities that include bovine milk, traditional beadwork, clothing, and accessories such as spears, walking sticks, and indigenous medicine (field observations, 2008–2015). Rural to urban migration may also contribute to the economy. These activities allow some access to food markets although quantities are limited and prices are high within the NCA compared to elsewhere in the region (field observations, 2008–2015). Restrictions on cattle grazing, and periodic bans on cultivation also contribute to persistent food insecurity in this community [41].

### Study design and participants

Study participants came from Endulen Village and surrounding areas within a few hours walking distance. With the assistance of 12 Maasai traditional birth attendants (TBA) we used chain-referral sampling to recruit 141 mothers and their infants (71 males and 70 females). The TBAs were drawn from a radius of 10 km around Endulen (S1 Fig). All births occurred at participants' homes during the dry season between June and September in 2010. Enlisting TBAs was critical to the success of this project. Access to newborn children is very limited by Maasai tradition and so measurement at birth by researchers would either not have been allowed or perceived as intrusive (field observations, 2008–2015). TBAs regularly provide care to expectant mothers and are present at birth in the majority of cases. Training TBAs to collect these data was the only method available to obtain the data necessary for this study. A further consideration is that Maasai households are dispersed over a large geographic area, and many households are accessible only by footpath. Maasai women rarely attend prenatal clinics, and instead seek the care of TBAs, who provide support during home deliveries. The TBAs were trained to collect data during multiple training sessions and focus groups [47]. They were equipped with data collection field kits that we developed according to community stakeholder input.

For baseline comparisons, we used WHO growth standards [48]. In addition, we analyzed a sample of (n = 102) neonates born at Bugando Medical Centre (Mwanza, Tanzania). These infants are of mixed socioeconomic background and from urban and peri-urban communities whose mothers attended antenatal clinics during the pregnancy period to monitor the progress

of the pregnancy. During the antenatal visits, they were given health education regarding nutrition. The information provided by healthcare practitioners does not report any specific diet restrictions among these pregnant women. Although our previous work has documented significant growth faltering among Mwanza children [49], the lack of evidence of specific third-trimester food restriction makes this sample suitable for comparative purposes. The inclusion of this cohort in this paper is not intended as a control as there are obviously many factors that distinguish them from the Maasai cohort. Birth outcome data are very sparse in Tanzanian and elsewhere in African low-income countries. For this reason, there is value in providing a comparative dataset from Tanzania to aid interpretation of the results of this study in addition to the WHO standards which serve as the baseline comparison.

## Ethics and privacy protection

Ethics approval was granted by the Conjoint Health Research Ethics Board (CHREB–Ethics ID: 23033), University of Calgary, and the National Institute of Medical Research (NIMR–Ethics ID: HQ/R.8A/Vol and HQ/R.8A/Vol I.107), Tanzania. The consent form was translated into KiSwahili and KiMaa. Because low literacy pervades the NCA, potential participants were verbally informed of the study details, and verbal consent was obtained by TBAs. Twelve TBAs were trained on the verbal consenting process over three training sessions. This verbal consenting process involving the TBAs was reviewed and approved by CHREB and NIMR.

All participants (TBAs, mothers and infants) were assigned a non-linkable identifier. Interviews data were securely stored on two laptop computers that were exclusively dedicated to the study. Field notes were delivered to a secure storage site upon return from the field. All data were transferred and converted to electronic format, and then securely stored at the Department of Cell Biology & Anatomy, University of Calgary. No information or data were disclosed to members outside the approved roster of study personnel.

## Quantifying maternal diet

To investigate maternal food intake throughout gestation, one of us (CP) first conducted a series of group interviews and generated observational field notes. Interviews were held with 12 TBAs who represented a cross-section of ages (30–50 years), were multiparous and had extensive knowledge of maternal practices. The initial group interview focused on the rationale for structuring and implementing a food frequency questionnaire. We recognized that the responses of the TBAs may have been influenced by social desirability. Maasai women may have been reluctant to disclose their actual dietary habits to avoid criticism if they did not conform the socially encourage practice of reducing food intake during the third trimester. To mitigate this potential bias, open-ended questions and informal dialogue to encourage active conversation were combined with participant-observation to confirm interview content. Using a refined version of the initial group interview, further interviews were done to confirm and elaborate on thematic content. Throughout these sessions, intersubjective meaning was established and conveyed as consensus by five or six participants on behalf of each group [50]. Validity and interpretation of data were verified through cross-case comparison, and continuous dialogue with TBAs about the topic of maternal health [51]. These data were transcribed and analyzed using NVIVO 9.

On the basis of the group interviews, we developed a food frequency questionnaire (FFQ) to measure maternal dietary intake, and to describe variation in maternal diet by measuring frequency and serving size. Due to the remote and isolated location of Maasai households and pervasive low literacy among participants, logistical challenges were met by developing the FFQ according to previously validated methods that rely on images of traditional food

containers [50,52–55]. Initial FFQ content was based on existing nutritional surveys of Maasai communities [20,56]. Food items were established through a series of group interviews with the 12 TBAs [51] based on open-ended questions about maternal diet. These discussions were interpreted in KiMaa, KiSwahili, and English by a community member who was experienced with implementing health research projects. TBAs identified food items commonly consumed during pregnancy. These items were recorded and compared to nutritional surveys reflecting typical maternal diets consisting of a narrow range of food items, and food consumption patterns of decline that mark the onset of third trimester food intake [20,56]. The initial FFQ was further reviewed by 36 women at various stages of pregnancy. These women independently confirmed the relevance of the initial list of items and suggested additional items for the second draft of the FFQ. The accuracy of the FFQ was improved by including habitual portion sizes [52,53]. Thumbnail photographs of FFQ items were presented to participants [55]. Items selected by the participants were pasted into the data collection booklet (S1 Appendix). Food items were measured using two 300 mL volumetric containers (one for dry food, and one for liquid food), which enabled participants to estimate serving sizes akin to those they regularly consumed.

Maternal dietary intake data were collected individually by the TBAs 2–3 days postpartum. Women were asked about their diet during early-mid pregnancy and during the third trimester and the TBAs filled out two FFQs per woman, one for each period. Given that the maternal diet is not altered until five to six months gestation, recall for early to mid-pregnancy was no longer than 3–4 months, which is an acceptable window for recall when using the FFQ method [57]. Moreover, because of seasonal food insecurity and limited subsistence options, the NCA diet is monotonous, which lends to reproducibility and accurate recall because fewer types of food items are consumed on a regular basis [58,59].

## Analysis of nutrition data

Tanzania Food Composition Tables (TFCT) were used for estimating the composition of all reported food items [60]. Compiled by the Harvard and Tanzanian Food and Nutrition Centre, the TFCT lists 47 nutrients and more than 400 commonly consumed items for the purpose of assessing links between nutrition and health outcomes [60]. We described caloric intake as kcal/day, and macronutrients (i.e., protein, fat, carbohydrates) as g/day. Since dietary restriction did not commence until trimester three, dietary intake from early to mid pregnancy is defined as "T1-2", and third trimester dietary intake is defined as "T3". Linear models were used to compare T1-2 and T3 dietary intake, in which time of gestation was included as a fixed factor, while TBAs and mothers were included as random factors to account for differences among TBAs in data collection and base-line differences among mothers. The models had random intercepts and random slopes to account for differences in the baseline as well as in the responses to the fixed factor. The significance of the differences in calorie and macronutrient intake between T1-2 and T3 was estimated by comparing these models against the null models by a likelihood ratio test using the ANOVA function in R. The mixed models were performed using the function lmer from the package lme4 for R and the p-values were obtained using the lmerTest package [61].

## Infant anthropometrics

Anthropometric data were gathered, by the TBAs, 48–72 hours postpartum. Using a portable medical hang-scale (Salter Breknell 235-S), BW was measured to 100 grams. Infants were wrapped in a cloth hammock and suspended from the anchored hang-scale. Hammock weight was subtracted from the measured BW. The HC (supraorbital ridge to occipital protuberance)

was measured twice to 0.5 cm on supine infants [62] using medical-grade tapes. The TBAs were trained by one of us (CP). To facilitate the measurement procedure and reduce the observer error, we provided images of the scale-face and paper tape in the booklet so that numeric values could be circled instead of hand-printed (S1 Appendix). A total of 140 Maasai infants were measured. Body weight and HC were compared to WHO growth standards for two-day-old infants. The WHO data are based on a large international sample of infants who were born into ideal socioeconomic conditions [63].

### Data analysis of anthropometric variables

An ANOVA test was performed first to assess the presence of systematic measurement error among TBAs. The linear model included the BW and HC as dependent variables, while TBA was set as the independent or fixed factor. The sample used for this analysis includes the 140 infants with anthropometric information. The results of these analyses showed a significant effect of TBA on both BW ($F_{(11, 128)} = 3.64$, $p<0.01$) and HC ($F_{(11, 128)} = 5.41$, $p<0.01$). Consequently, the variation associated with TBAs, was removed by standardizing all values to the grand TBA mean (mean of the TBA means) for each variable. This removes systematic error due to differences among TBAs in data collection. We checked the TBA standardized data for outliers and eliminated all values that fell four or more standard deviations (SD) from the mean. For our sample size, a cut-off point of 4 SD for outliers was recommended based on simulation studies [64]. Infants with no sex information were also excluded from the subsequent analyses due to the dependence of z-values estimations on this variable. After elimination of outliers and missing data, the NCA sample for anthropometric analysis was composed of 116 individuals. To assess whether the z-scores for HC and BW of the NCA sample differ from the WHO reference, we used a one sample T-test.

In order to account for the allometric relationship between HC and BW, the expected HC given the BW was estimated for infants from the two Tanzanian samples on the basis of values derived from the WHO reference. A linear regression was used to describe the relation between both variables and the parameters (intercept and slope) obtained from this analysis were then applied to the estimation of HC from BW. For the NCA sample, the estimations were based on the values adjusted by TBA expressed in cm and grams, respectively.

## Results

### Maternal diet during pregnancy

During group interviews, participants reported the dietary restriction towards the third trimester of gestation. The diet of Maasai women during T1-2 remained the same as it was prior to pregnancy. The most common foods reported by participants included maize porridge, white rice, corn, beans, cooked cabbage, milk, and liquefied goat or cow fat. At the beginning of the third trimester, foods described as "dangerous" (e.g., fresh milk, moderate cooked meat) were replaced by foods described as "safe" (e.g., sour milk, animal fat), and the quantities of all food types were significantly reduced. In addition, participants reported using herbal medicines with emetic properties. From the interviews, it also emerged that domestic workloads increased over the initial four months of gestation. Although not quantified in this study, it is likely that this would create an increase in calorie expenditure. A summary of the main statements that emerged in the interviews is provided in S2 Appendix.

Analysis of the FFQ showed that Maasai women changed their dietary intake at specific gestational time-points (Fig 1). The results reported here are based on 68 records for the first two trimesters and 79 for the third trimester, after outliers and missing data were removed. At the onset of third trimester, daily calorie consumption was reduced from a mean of 1601 (± 734.19

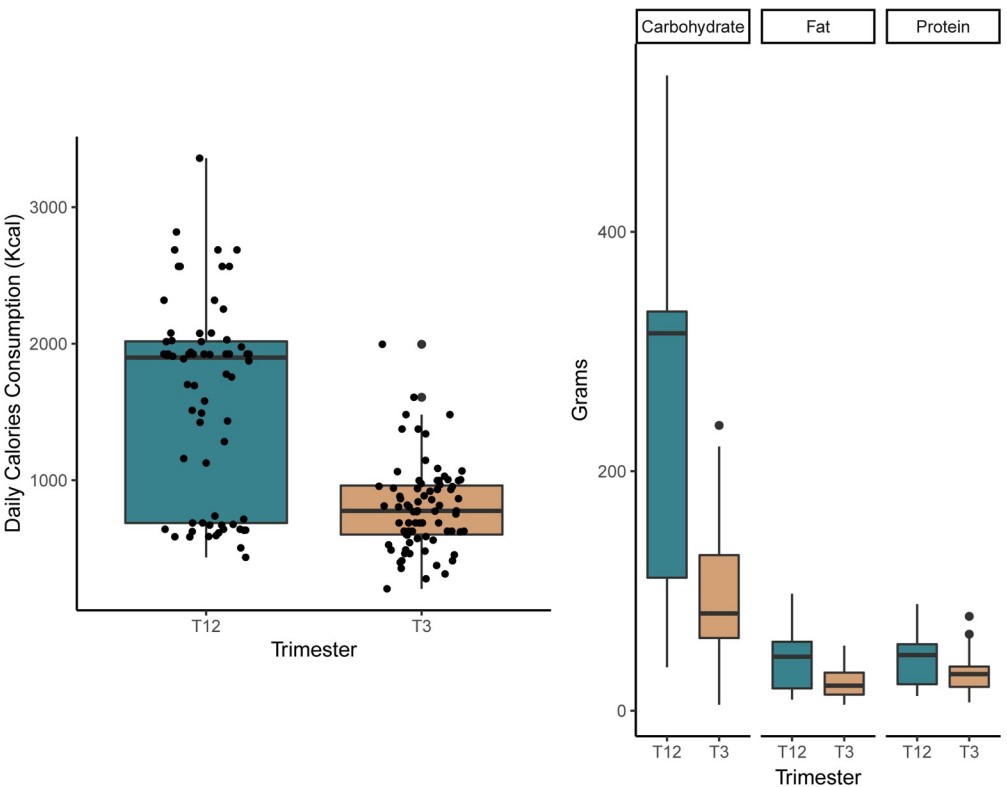

**Fig 1. Maternal calorie intake and diet composition during the first two trimesters (T12) and the third trimester (T3).**

SD) kcal/day (T1-2) to 799 (± 317.59 SD) kcal/day. Across TBAs some participants appeared to ingest significantly fewer calories than average (S2 Fig). Since it was not possible to determine whether these differences were due to systematic measurement error or real variation in food intake among participants, the variation potentially due to TBAs was incorporated as a random factor into the statistical models. The results show that there was a significant reduction in calories and in the amount of the three macronutrients from early-mid pregnancy to the third trimester (Table 1). The least squares means estimated from these models show that during the first two trimesters, pregnant women consumed 276.04 g/day carbohydrates (95% CI: 237.72–314.37), 43.83 g/day fat (37.67–49.99) and 45.27 g/day protein (8.69–51.86). At the onset of T3 these values dropped to 100.27 g/day (62.46–138.08), 23.43 g/day (17.38–29.48) and 30.17 g/day (23.69–36.65), respectively. The greatest proportion of nutrient reduction during third trimester was in carbohydrates, followed by fat and protein (Table 1).

**Table 1. Results of linear mixed models analyses by dietary component to estimate the effect of gestation period (fixed factor or independent variable) on diet composition (dependent variables).** In these models, TBAs and mothers were included as random factors. The negative sign of the slope (b) indicates a reduction of each dietary component from early-mid to late pregnancy.

| Diet Component | Estimate (b) | Standard error | F | p |
|---|---|---|---|---|
| Calorie | -902.35 | 74.94 | 146 | <0.01 |
| Protein | -15.099 | 2.47 | 37.443 | <0.01 |
| Fat | -20.397 | 2.32 | 77.534 | <0.01 |
| Carbohydrate | -175.78 | 13.14 | 179.05 | <0.01 |

**Table 2. Means and standard deviation (SD) for weight and head circumference of infants (females = F and males = M) from Ngorongoro Conservation Area (NCA) and Bugando Hospital samples.**

| Sample | N | Sex | Weight (g) | Head Circumference (cm) |
|---|---|---|---|---|
| | | | Mean ± SD | Mean ± SD |
| NCA | 60 | F | 2976.10 ± 554.24 | 31.94 ± 1.69 |
| | 56 | M | 2848.78± 536.99 | 31.97 ±1.49 |
| Bugando | 58 | F | 2974.48 ± 493.22 | 34.15 ± 1.28 |
| | 44 | M | 3116.82 ± 506.23 | 35.01 ± 1.24 |

We then assessed the association between reduced food intake on newborn size estimating linear models with weight and HC as dependent variables, the amount of reduction in diet components (calories, protein, carbohydrate, fat) between early-mid and late pregnancy as fixed effects, and TBA and mother as random effects. A significant relationship between infant weight and calorie and protein reduction was found but not for HC (S1 Table). These results need to be taken with caution. Because all Masaai women in the study reduced their food intake, the range of variation of the differences between T1-2 and T3 is relatively small. If a dose-response relationship is present, it would require a larger sample size to be detected.

### Infant weight and head circumference

Descriptive statistics for BW and HC in the NCA and Bugando samples are provided in Table 2. Z-scores for BW and HC (Fig 2) of the NCA sample differed significantly from the WHO standard (BW: df = 115, T = -6.8, $p < 0.0001$; HC: df = 115, T = -14.6, $p < 0.0001$).

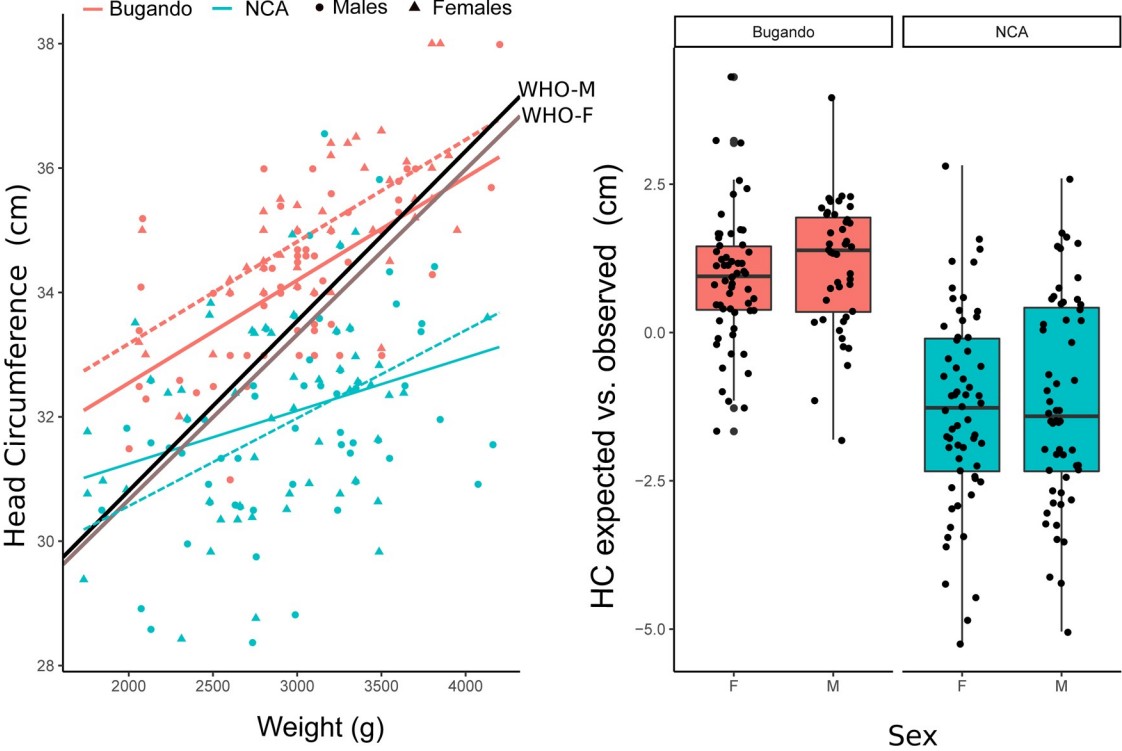

**Fig 2. Z-scores of weight and head circumference of female (F) and male (M) newborns from Ngorongoro Conservation Area (NCA) and Bugando Hospital samples.** Dashed line: males.

Maasai males departed from the standards in weight more than females (df = 115, F = 5.5, $p$ = 0.02). Notably, HC values for both sexes fell further below the WHO standard than did BW values (Fig 2). The average value for Maasai HC was remarkably low, falling 1.7 standard deviations below the WHO standard. Thus, compared to the standard, HC was more markedly reduced than weight (df = 115, T = 1.9, $p$ < 0.0001).

Fig 2 shows a scatter plot of the WHO Z-score for HC plotted against the Maasai Z-score for BW. The correlation between HC and BW values was 0.41, which is within the range found in other studies [65]. Fig 2 also shows the proportion of the distribution that falls within the microcephalic (< 2 SD) and low birth weight (LBW< 2500 g) ranges. Thirty-one percent of

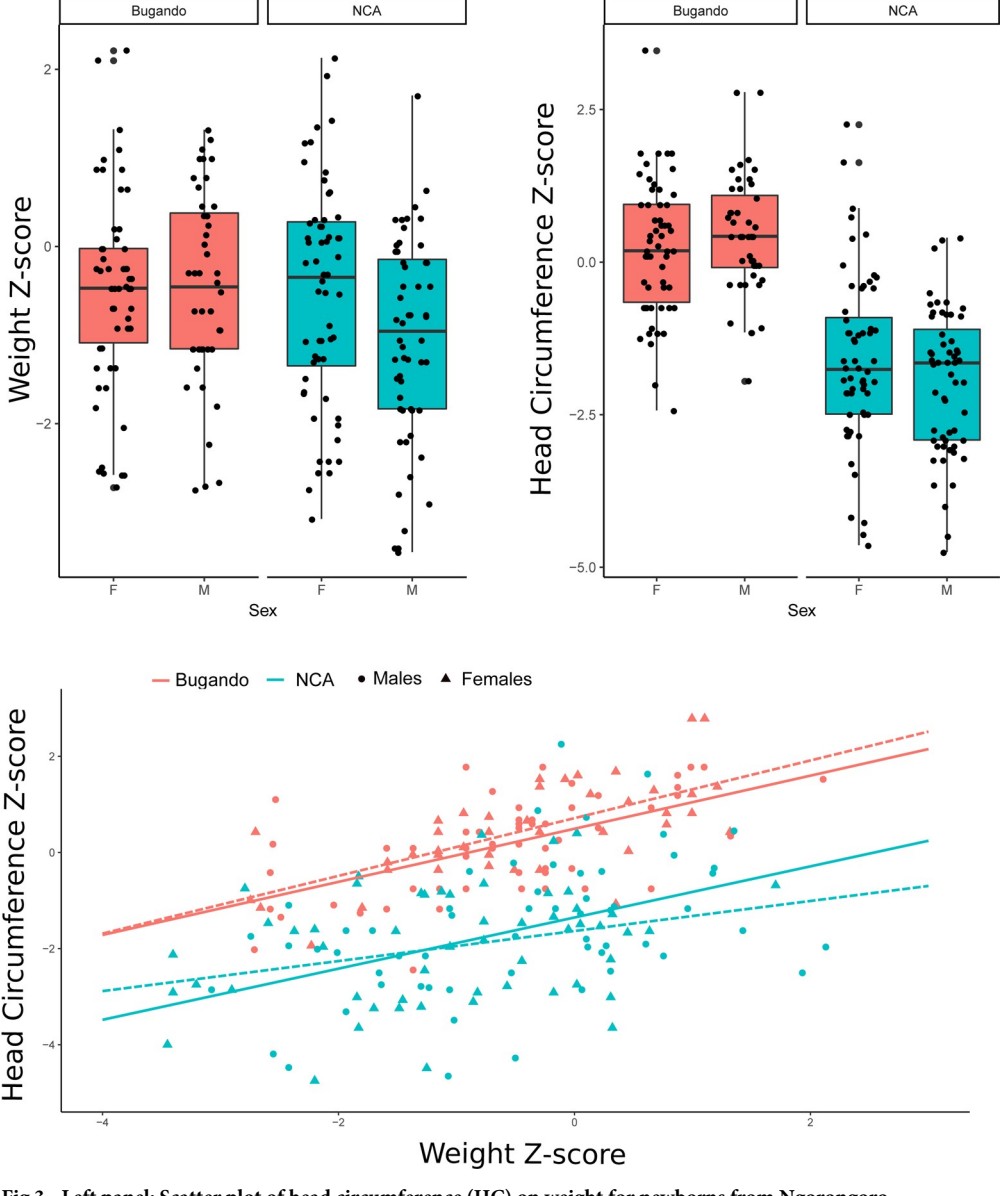

**Fig 3. Left panel: Scatter plot of head circumference (HC) on weight for newborns from Ngorongoro Conservation Area (NCA) and Bugando samples.** Lines represent the adjusted linear model by sample and sex (Dashed line = males). For comparison, the expected regression lines for the WHO standard are also depicted. Right panel: Box plots showing the distribution of differences between observed values of the HC in Bugando and NCA samples with the values estimated using the regression lines from the WHO standard. F: female; M: male.

the Maasai sample exhibited LBW, and 40% of infants were microcephalic. Thirty-four percent of infants fell below the second percentile of the WHO standard for HC, and 16.5% of infants exhibited LBW and microcephaly. These values contrasted with those obtained for the sample of Bugando Hospital, in which 12% of the infants exhibit LBW and only two (of 102) infants had HC Z-scores lower than 2 (Fig 2).

Using the WHO reference data, we predicted HC for BW for Maasai and Bugando samples. Both for males and females the observed values in the Maasai sample were smaller than expected, while the HC of neonates from Bugando Hospital were, on average, larger than estimated based on the WHO reference data (Fig 3).

## Discussion

This is the first study to examine the quantitative link between third trimester dietary restriction and birth outcomes in humans. We found a significant reduction in third trimester food intake among NCA Maasai women using a food frequency questionnaire. These findings are consistent with previous studies concerning maternal practices among Kenyan and Tanzanian Maasai [20,56]. Weight at birth and HC of Maasai infants were dramatically below the WHO standard, with HC more severely affected than weight. Maasai infant birth weight and head circumference are also significantly lower than those obtained from another Tanzanian cohort in which there is less evidence of malnutrition and no reports of dietary restriction during pregnancy.

We estimate that during the first two trimesters, pregnant women in the NCA consumed an average of 1601 kcal/day. During the third trimester, caloric intake was further reduced to 799 kcal/day. Analysis of dietary components revealed that the largest reduction was in carbohydrates, followed by fat and protein. In the NCA currently, imported maize is a dietary staple and the major source of carbohydrates. Accordingly, our analysis of the food frequency questionnaire data confirms the important role of maize in the maternal diet. In the first and second trimesters, 76% of total energy came from carbohydrates while this was reduced to 64% in the third trimester.

Throughout pregnancy, Maasai women remain physically active. Domestic workloads typically include strenuous walking for up to 10 km per day, as well as non-mechanized labour such as carrying 10 to 20 litres (~12 to 25 kg) of water and collecting 20 to 25 kg of firewood twice per week. The FAO [6] (2001) estimates that these activities require daily energy intake of 2,500 to 2,650 kcal, and a well-nourished pregnancy requires an additional 300 kcal/day for prenatal tissue synthesis [7]. Thus, NCA Maasai women consume between 700 and 850 fewer kcal/day than are recommended, even before pregnancy and during the first two trimesters. During the third trimester food restriction, energy intake is deficient by 1,900 to 2,050 kcal/day. While this energy deficit is remarkable, it is not unique. In The Gambia, research on maternal diet found a similar trend in food consumption among rural women [66,67].

The low body weight of Maasai infants compared to WHO standards is not surprising. However, the lack of evidence for head sparing is remarkable. Growth restriction typically affects body mass more than head size in both rodent models [30] and humans [28,68]. One possible explanation for the absence of head sparing is that there is a genetic basis for small HC at birth in the NCA Maasai population. This possibility is extremely unlikely. Although data on growth in Sub-Saharan Africa under ideal nutritional conditions is very limited, WHO standards are intended for population comparisons. The rationale for global use of these standards is that 85–90% of the total genetic variance for stature or infant length exists within populations and only a small percentage is attributed to differences among populations [68–71]. Nevertheless, genetic differences may explain some deviations from the WHO standards in

particular populations. A recent analysis of worldwide variation in child growth data suggested that mean HC can vary up to 05–1.0 SD from the WHO standards [72]. At 1.7 SD below the WHO mean, however, the NCA infant data are outside this range of variation. A second alternative explanation is that the magnitude of nutritional stress experienced by Maasai mothers is so severe that brain sparing does not occur. Support for this possibility comes from studies among aboriginal Australians and Nepali children in which the both communities experienced severe nutritional stress. In both cases, approximately 50% of HC Z-scores were $<-2$, meeting criteria for the diagnosis of microcephaly [5,73]. Finally, the absence of head sparing and high incidence of low HC values at birth among Maasai might also be due to the timing of nutritional stress in the third trimester—a critical window in which 65% of fetal brain volume is accumulated [74]. This factor might be exacerbated by the background level of nutritional stress evident in the NCA Maasai population.

The timing of maternal food restriction during pregnancy is known to affect birth outcomes. Studies of the Dutch famine cohort reveal that metabolic and cardiovascular disease effects were most severe for those affected in early and mid- pregnancy compared to those affected only in the third trimester [75]. However, studies of the Generation R cohort in Holland, born between 2002 and 2006, show that slowed fetal growth in the third trimester is associated with adverse neurodevelopmental outcomes [33,76]. This latter finding is congruent with imaging studies demonstrating associations between low weight at birth and altered white matter integrity [77]. To our knowledge, there are no experimental or epidemiological studies that specifically address the scenario of dramatic food restriction during the third trimester in humans.

Head circumference is a known etiological factor in birth complications [78], particularly relative to maternal size [79], and is at the heart of the obstetric dilemma [24]. The practice of third trimester nutrient reduction represents a desire to avoid the tradeoffs implicit in the obstetric dilemma. Our results show reduced HC at birth. We do not know whether this achieves the desired outcome of safer childbirth. However, reduced birth-weight and HC are associated with deleterious outcomes for children, including neurodevelopmental delay [80,81], increased risk of cardiovascular disease [82,83], metabolic syndrome [84,85] and infectious disease risk [86]. Any reduced risk of birth complications thus comes at the cost of significantly increased later health risks for the offspring.

There are several caveats to consider for interpreting our results. The FFQ method has limitations, particularly for quantitative estimates of food intake [50]. We addressed this by asking participants to demonstrate typical serving sizes of known volume. Further, the dietary patterns reported are complex, particularly when considering the possible effects of emetics or other herbal remedies that are not considered in the FFQ-based caloric estimates. Several studies have indicated that long-term quantitative recall allows for comparing general trends in food consumption patterns [87,88]. The limitations of the food survey data mentioned above make the significant values and the consistent reduction of food intake across TBAs all the more remarkable.

A second, related, caveat is the potential for bias in data recording. The TBAs conducted the FFQ interviews after the birth and after taking the anthropometric measurements. While they would not have had access to WHO standards, it is still possible that the condition of the infant influenced the conduct of the FFQs. The fact that we, perhaps surprisingly, observed only very low correlations between the measurements and the FFQ data makes it unlikely that this factor, if present, biased our results. This is, however, a factor that should be considered in interpretation of these data.

A third limitation relates to gestational age. Maasai women approximate gestational age by using consistent calendar events such as market days to monitor their menstrual cycles. Even though we cannot discard this factor, differences in gestational age among newborns seem

unlikely to significantly undermine our overall findings. Mean HC for the NCA sample is below the 50th percentile at 36 weeks of gestation according to estimations from a multinational longitudinal study [89]. Conversely, weight falls above the 50th percentile, which reinforces the conclusion that fetal growth restriction in this cohort occurs without head sparing. Finally, although we are confident in the majority of measurements taken by the TBAs, measurement error for both birth weight and head circumference are likely higher in this study than for those based on data collected by trained researchers. For this reason, we carefully considered variation among TBAs and eliminated outliers as described in materials and methods. Most outliers were for very low values and so this would not have shifted the results towards the conclusions reached. None of the data that we report here could have been obtained without the willing and generous participation of volunteer traditional birth attendants.

Our finding of alarming rates of low head circumference and birth weight among Maasai women raises the obvious question of intervention to improve birth outcomes in this vulnerable population. Here, the engagement of traditional birth attendants in this work is crucial. The cultural context of the practices in question are likely long-standing. The extent to which their effects are exacerbated by more recent food insecurity is unknown. However, development of culturally appropriate intervention strategies must engage Maasai community stakeholders including both Maasai women and the traditional birth attendants who care for them and their infants. The engagement of TBAs in the design and conduct of this study is a small but important step in that direction.

In conclusion, NCA Maasai women dramatically reduce food intake during the third trimester. This practice is associated with low birth weight and disproportionately reduced head circumference. These birth outcomes are associated with health effects that likely exacerbate the health risks that affect this population. These findings also raise important questions about the determinants of head sparing with low birth weight that merit further study in both animal models and in humans. Finally, this study also highlights the need to take into account cultural practices as those described here, which can reduce the effectiveness of food-based nutritional interventions if not addressed properly. More effort is required to implement interventions for maternal undernutrition because of their potential to improve outcomes for maternal and infant health. Such interventions are complex and difficult, but undertaking them is crucial for achieving meaningful improvements in the health and well-being of vulnerable populations in many areas of the world.

## Supporting information

**S1 Fig. Map of Northern Tanzania and Southern Kenya showing the location of the NCA (dark shaded).** Data collection was centered in Endulen, but subjects were enrolled from within the entire NCA. Adapted from Coast (2001).
(DOCX)

**S2 Fig. Daily calorie consumption recorded by each traditional birth attendant (TBA).**
(PDF)

**S1 Table. Linear mixed model with weight and HC as dependent variables, the amount of reduction in diet components (calories, protein, carbohydrate, fat) as fixed effects, and TBA and mother as randon effects.**
(DOCX)

**S1 Appendix. Food Frequency Questionnaire and data collection booklet.**
(DOCX)

**S2 Appendix. Summary of the main statements emerged in the interviews.**
(DOCX)

## Acknowledgments

We are grateful to the Maasai women enrolled in this study for sharing their time and experience with the investigators, as well as to the Maasai community at Endulen for their hospitality and for welcoming the researchers who participated in this study. We thank the staff at Endulen Hospital and the Catholic Archdiocese of Arusha for hospitality and support of our work in the NCA. We thank Jennifer Hatfield for her leadership of the University of Calgary Global Health initiative in Tanzania and for advice on many aspects of this project. Kimani Kangweli, Emmanuel Shangai, Salehe Mganzila, Nicola Hahn, Kai Lukowiak, and Tegan Barry contributed to logistics and anthropometric data collection for this project. The work of the traditional birth attendants was essential for this study. We thank the Traditional Birth Attendants for their invaluable work on this project. This study would not have been possible without their invaluable advice and contributions to data collection.

## Author Contributions

**Conceptualization:** Christopher D. Powell, Warren M. Wilson, James C. Cross, Kenneth Lukowiak, Benedikt Hallgrimsson, Paula N. Gonzalez.

**Data curation:** Christopher D. Powell, Paula N. Gonzalez.

**Formal analysis:** Christopher D. Powell, Benedikt Hallgrimsson, Paula N. Gonzalez.

**Funding acquisition:** Richard Spritz, Benedikt Hallgrimsson.

**Investigation:** Godwin Olesaningo, Mange Manyama, Richard Spritz, Kenneth Lukowiak, Benedikt Hallgrimsson.

**Methodology:** Paula N. Gonzalez.

**Project administration:** Christopher D. Powell, Heather Jamniczky, Kenneth Lukowiak, Benedikt Hallgrimsson.

**Resources:** Benedikt Hallgrimsson.

**Supervision:** Warren M. Wilson, Heather Jamniczky, Benedikt Hallgrimsson.

**Writing – original draft:** Christopher D. Powell, Warren M. Wilson, Godwin Olesaningo, Mange Manyama, Heather Jamniczky, Richard Spritz, Kenneth Lukowiak, Benedikt Hallgrimsson, Paula N. Gonzalez.

**Writing – review & editing:** Christopher D. Powell, James C. Cross, Kenneth Lukowiak, Benedikt Hallgrimsson, Paula N. Gonzalez.

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
