## [Decision Letter · Decision Letter 0]

5 Dec 2019

PONE-D-19-25660

Dietary practices during pregnancy among Tanzanian Maasai and its effect on growth status at birth

PLOS ONE

Dear Dr Paula Gonzales ,

Thank you for submitting your manuscript to PLOS ONE. After careful consideration, we feel that it has merit but does not fully meet PLOS ONE’s publication criteria as it currently stands. Therefore, we invite you to submit a revised version of the manuscript that addresses the points raised during the review process.

ACADEMIC EDITOR: Please insert comments here and delete this placeholder text when finished. Be sure to:

Indicate which changes are required versus recommended for acceptanceAddress any conflicts between the reviewsProvide specific feedback from your evaluation of the manuscript

We would appreciate receiving your revised manuscript by Jan 19 2020 11:59PM. To enhance the reproducibility of your results, we recommend that if applicable you deposit your laboratory protocols in protocols.io, where a protocol can be assigned its own identifier (DOI) such that it can be cited independently in the future. For instructions see: http://journals.plos.org/plosone/s/submission-guidelines#loc-laboratory-protocols

We look forward to receiving your revised manuscript.

Kind regards,

Rosely Sichieri

Academic Editor

PLOS ONE

Journal Requirements:

3. We note that Figure [2] includes an image of a [patient / participant / in the study]. 

Additional Editor Comments (if provided):

A food frequency questionnaire (FFQ) was developed and used. The easiest way to get food intake from each trimester would be a few 24 h recall. Why authors choose the FFQ?

It is not clear in the text the frame time for the FFQ neither how many and when the FFQ was obtained. The text reads 2-3 days postpartum and a time frame of 3-4 months?

In results “The diet of Maasai women during T1-2 remained the same as it was prior to pregnancy”, but FFQ was reported for pre-pregnancy? It is quite confuse the points in time of FFQ.

In order to compare trimesters, dietary intake during trimesters one and two were combined? How the mean? This is not appropriated. Changes over the three trimesters of dietary intake would be the adequate approach.

How many Infants had no sex information?

The outliers plus no sex information is about 20% a large amount.

It is not clear the need to compare the expected HC given the BW for the infants from the two Tanzanian samples based of the values derived from the WHO reference. Comparison between the two samples would be easily interpreted.

Fig 1 and 2 could be moved to supplementary data.

In figure 3- If T1 and T2 were averaged we expected less variation compared to T3.

Fig 5- authors could keep only the figure with the curves, data on the right is less informative.

Reviewers' comments:

Reviewer's Responses to Questions

**Comments to the Author**

1. Is the manuscript technically sound, and do the data support the conclusions?

Reviewer #1: Yes

2. Has the statistical analysis been performed appropriately and rigorously? 

Reviewer #1: Yes

3. Have the authors made all data underlying the findings in their manuscript fully available?

Reviewer #1: Yes

4. Is the manuscript presented in an intelligible fashion and written in standard English?

Reviewer #1: Yes

5. Review Comments to the Author

Reviewer #1: Comments to the author

This is an original article that contributes greatly to a specific knowledge gap. The manuscript is well well written and and I considered the discussion section quite complete. However, I have a few points that I suggest to be clarified in the text.

- Was the food consumption of the Bugando sample not assessed? How to ensure that there was no decrease in the food consumption of some mothers, given the variety in the socioeconomic context, as mentioned in the text? I consider that it is not sufficient to state that “there is no evidence of specific third-trimester food restriction in that population”.

- Line 347 – “it was after accounting for TBA and maternal effects” – What factors were these considered? The title of table 1 also needs to be improved and incorporated into the footer which are the adjustment variables.

- I suggest further analysis on association between reduced food intake (difference between T1-2 and T3 - continuous or tertile) on newborn size, controlled for "TBA and maternal effects”. Perhaps this association could reinforce causality by showing a dose-response relationship.

- In general, I suggest that the titles of the figures and tables be revised. The information of who and when should be added.

- Table 2: after the +/- sign would be the SD? The information should be included in the header line of the table. Also, information about female (F) and male (M) should be included in the table title.

6. PLOS authors have the option to publish the peer review history of their article (what does this mean?). If published, this will include your full peer review and any attached files.

Reviewer #1: No

---

## [Author Response · Author response to Decision Letter 0]

31 Jan 2020

We thank the reviewer and the Editor for their comments which helped to improve our manuscript. We have addressed and incorporated the suggestions made by the reviewers and the Editor. 

Editor Comments

Comment 1: A food frequency questionnaire (FFQ) was developed and used. The easiest

way to get food intake from each trimester would be a few 24 h recall. Why authors choose the FFQ?

Reply: The different methods for dietary assessment proposed in the literature all have strengths and limitations (see review in Shim et al., 2014, Epidemiology and Health, 36). The selection of one specific method will depend on the aims of the study and the feasibility according to the human and material resources available. The 24h recall is a subjective measure based on open-ended questionnaires administered by a trained interviewer. Its main adjvantage is that it provides detailed intake data and no literacy is required. One limitation for the purpose of our study is that the method mainly focuses on short-term intake, while we were interested in long-term dietary exposure. This means that multiple interviews would be needed to assess food intake. This is both an expensive and time-consuming. Based on our resources this was deemed to be inadequate for our study in Tanzania. Additionally, a possible negative consequence of this method is that it can result in changes to diet as a result of repeated interviews. 

On the other hand, the FFQ method estimates food intake over relatively longer periods. It is both cost-effective and time-saving, and thus suitable for epidemiological studies (Jackson et al., 2012, Public Health Nutrition: 16; Zack et al., 2018, Public Health Nutr. 21). Since the FFQ is specific to each study group; we carrefully designed and validated the questionnaire before gathering data on dietary intake among Maasai women.

Briefly, a group interview was first held with a group of traditional birth attendants and mothers. Participants represented a cross-section of ages (30-50 years), were multiparous and were familiar with dietary restriction. The initial group interview focused on the rationale for structuring and implementing dietary survey. Then, a workshop was held with the TBAs, who were asked to list food items that are commonly consumed during pregnancy. These items were recorded and compared to previous studies on maternal diets of Kenya Maasai (Nestel 1989; Mpoke 1993). Upon confirming the relevance of our initial food list, we developed a first draft of the FFQ which was disseminated to the TBA group interview participants, and was reviewed by randomly selected women at various stages of pregnancy (N=36). These women confirmed the relevance of the initial food item list, and suggested additional items for the FFQ, which was distributed to the TBA field team. The TBAs had occasion to practice interview, and provided feedback about how to improve data accuracy. Then, we held a training session to review data collection techniques and the TBAs proceeded to the data collection phase.

Comment 2: It is not clear in the text the frame time for the FFQ neither how many and when the FFQ was obtained. The text reads 2-3 days postpartum and a time frame of 3-4 months?

Reply: Maternal data were collected two to three days postpartum. During the interview, the TBAs asked about the dietary intake during the first two trimesters and the last trimester, filling out a FFQ for each period. 

Since maternal diet is not altered until five to six months into pregnancy, recall for early to mid pregnancy was no longer than three to four months, which is an acceptable window for recall when using the FFQ method (Buzzard et al., 2001; Michels et al., 2005). Moreover, because of seasonal food scarcity and limited subsistence options, the NCA diet is monotonous, which lends to reproducibility and accurate recall because fewer types of food items are consumed on a regular basis (Beaton et al., 1979; Kigutha, 1997). A total of 68 FFQs for the first two trimesters and 79 for the third trimester were included in this study. We incorporated this information in the Results section of the new version of the manuscript. 

Comment 3:In results “The diet of Maasai women during T1-2 remained the same as it was prior to pregnancy”, but FFQ was reported for pre-pregnancy? It is quite confuse the points in time of FFQ. In order to compare trimesters, dietary intake during trimesters one and two were combined? How the mean? This is not appropriated. Changes over the three trimesters of dietary intake would be the adequate approach.

Reply: In the reply to the first comment we provided more details about the methods for developing the food frequency questionnaire. Before implementing the FFQ, one of us (CP) had group interviews with TBAs and mothers in which they were asked about their dietary habits. From these interviews it was established that dietary changes occur later in pregnancy. This is in agreement with findings for other Maasai groups. On this basis, we decided to focus the FFQ on dietary intake during two clearly differentiable periods of pregnancy, before and after the pregnant women start the dietary restriction. We agree with the reviewer that it would be ideal to gather data throughout gestation but this was not feasible because of our limited resources and the difficulties for access both to the study area and the participants. Finally, as mentioned above the diet is very monotonous. 

Comment 4: How many Infants had no sex information? The outliers plus no sex information is about 20% a large amount.

Reply: The sample of Maasai infants with anthropometric data was composed by 140 individuals, from which 24 had no sex information. Consequently, the estimation of z-scores and further analyses based on these scores were performed using the sample of 116 infants with sex information. 

Comment 5: It is not clear the need to compare the expected HC given the BW for the infants from the two Tanzanian samples based of the values derived from the WHO reference. Comparison between the two samples would be easily interpreted.

Reply: The reviewer is correct that a comparison between the two simples can easily be interpreted. However, the analysis as a whole uses the WHO reference as the baseline comparison. It is, therefore, appropriate to determine deviataion from the expected relationship between head circumference and body mass using this reference and to determine how the two samples lie with respect to the WHO sample. There is a statistical advantage to this as the WHO sample is so large that the relationship can be assumed to be measured without error. Further, the point of the comparison to the Bugando sample is simply to show that we do not obtain similar results in another Tanzanian sample in which we have no evidence of third trimester food restriction. It is not to use the Bugando sample as an appropriate baseline comparison. Had we done the comparison as the reviewer suggests, our study could be criticized on the basis of the appropriateness of the Bugando sample as an “optimal nutrition baseline” which it is clearly not.

Comment 7: Fig 1 and 2 could be moved to supplementary data.

Reply: Figure 1 was moved to supplementary data, while figure 2 was eliminated from the new version of the manuscript.

Comment 8: In figure 3- If T1 and T2 were averaged we expected less variation compared to T3.

Reply: The larger variation for the first period suggests the existence of individual differences in the food intake which are reduced by late gestation, when dietary intake is more regulated by cultural practices. Given this, it is not unexpected to find more variation for T1-T2. Also, it should be noted that, as we explain above, information about food composition for early-middle pregnancy was not obtained separately for each trimester. Consequently, the values in Figure 3 do not represent an average. 

Comment 9: Fig 5- authors could keep only the figure with the curves, data on the right is less informative.

Reply: These figures both contain Information necessary to understand the paper. We have addressed this by better explaining these figures in the text and captions.

Reviewers' comments:

Reviewer #1: Comments to the author

Comment 1: Was the food consumption of the Bugando sample not assessed? How to ensure that there was no decrease in the food consumption of some mothers, given the variety in the socioeconomic context, as mentioned in the text? I consider that it is not sufficient to state that “there is no evidence of specific third-trimester food restriction in that population”.

Reply: The Bugando sample was incorporated to compare the anthropometric measurements at birth with a Tanzanian population in which the cultural practice of third trimester restriction is not extended. The women included in this sample attended antenatal clinics during the pregnancy period to monitor the progress of the pregnancy. During the antenatal visits, they were given health education regarding nutrition. The information provided by healthcare practitioners does not report any specific diet restrictions among these pregnant women, although food consumption was not assessed in Bugando sample because it was not part of the original research project. Therefore, even though we can not discard variation in food intake due to socio-economic status and availability of food, we consider this is an appropriate reference for comparative purposes. 

Comment 2:- Line 347 – “it was after accounting for TBA and maternal effects” What factors were these considered? The title of table 1 also needs to be improved and incorporated into the footer which are the adjustment variables.

Reply: We modified the title and provided more information about the variables used in these models.

Comment 3: I suggest further analysis on association between reduced food intake (difference between T1-2 and T3 - continuous or tertile) on newborn size, controlled for "TBA and maternal effects”. Perhaps this association could reinforce causality by showing a dose-response relationship.

Reply: Following reviewer’s suggestion, we estimated linear models with weight and HC as dependent variables, the amount of reduction in diet components (calories, protein, carbohydrate, fat) as fixed effects, and TBA and mother as randon effects. We found a significant relationship between infant weight and calorie and protein reduction, while the effect was non significant for HC. These results need to be taken with caution. Because all Masaai women reduced their food intake, the range of variation of the differences between T1-2 and T3 is relatively small, and thus even if a dose-response relationship is present it would require a larger sample size to be detected. The results from this analysis were included in a table as Supporting Information (S5 Table). 

Comment 4: In general, I suggest that the titles of the figures and tables be revised. The information of who and when should be added.

Table 2: after the +/- sign would be the SD? The information should be included in the header line of the table. Also, information about female (F) and male (M) should be included in the table title.

Reply: The titles of the tables and figures were revised according to the reviewer’s suggestion.

---

## [Decision Letter · Decision Letter 1]

22 Jun 2020

PONE-D-19-25660R1

Dietary practices during pregnancy among Tanzanian Maasai and its effect on growth status at birth

PLOS ONE

Dear Dr. Gonzalez,

Thank you for submitting your manuscript to PLOS ONE. After careful consideration, we feel that it has merit but does not fully meet PLOS ONE’s publication criteria as it currently stands. Therefore, we invite you to submit a revised version of the manuscript that addresses the points raised during the review process.

We look forward to receiving your revised manuscript.

Kind regards,

Jai K Das

Academic Editor

PLOS ONE

Reviewers' comments:

Reviewer's Responses to Questions

**Comments to the Author**

1. If the authors have adequately addressed your comments raised in a previous round of review and you feel that this manuscript is now acceptable for publication, you may indicate that here to bypass the “Comments to the Author” section, enter your conflict of interest statement in the “Confidential to Editor” section, and submit your "Accept" recommendation.

Reviewer #2: (No Response)

Reviewer #3: (No Response)

2. Is the manuscript technically sound, and do the data support the conclusions?

Reviewer #2: Partly

Reviewer #3: Partly

3. Has the statistical analysis been performed appropriately and rigorously? 

Reviewer #2: Yes

Reviewer #3: Yes

4. Have the authors made all data underlying the findings in their manuscript fully available?

Reviewer #2: Yes

Reviewer #3: Yes

5. Is the manuscript presented in an intelligible fashion and written in standard English?

Reviewer #2: Yes

Reviewer #3: Yes

6. Review Comments to the Author

Reviewer #2: There are significant editorial issues remaining - ranging from grammatical to spelling. I would strongly suggest that you take time to have the document propoerly edited. Spacing errors and puncttuation errors must be atteded to. Please use the same font size throughout the document (including the reference list).

I struggled with your comparison to the Bugando maternal-child dyads - these are non-Maasi peoples with very distinct 'tribal' variances that you have not corrected for. I am not convinced you actually can make this a reasonable comparison. Even the urban/per-urban factor was a further confusion, as the NCA residents are rural pastoralists. i remained confused as to why you did this component of the study given the minimal data collected.

your consnet process also remains questionable. I would want more details about the TBAs role as my knowledge of this population (where I have done many studies) would indicate a concern with the TBAs literacy. So if you are asking them to read the consents to the participants this is highly unlikely from my experience. I would stress that you explain what was actually done.

The dietary recoall for 9 months is much more is a high expectations. The dietary intake is only one aspect of the nutrition in this gorup - for example, herbs to cause emesis(vomitting) and what quality of food is actually allocated has not been addressed in your paper. I am concerned about this possible misrepresenation.

Reviewer #3: Interesting and well written report on the association between third trimester maternal food intake restriction and birth sizes. The report is transparent with regard to the methodologies applied and possible alternative explanations, but there are some issues to address further.

Authors rightfully discuss social desirability as a possible source of bias with TBAS’s. However, could there also be another source of bias? TBA’s collected maternal dietary data 2-3 days postpartum, presumably having already measured birth sizes. How exactly do socially desirable answers play a role once the child was already born alive and presumably healthy albeit (too) small? A second question is if the knowledge of birth sizes could have influenced measurements through FFQ’s? Or, for that matter, have there been occasions where FFQ’s were taken before birth size measurements, where knowledge of birth sizes may have influenced the actual birth measurements? If this kind of information bias, which cannot be analytically solved, could have occurred it should be briefly discussed in the limitations section.

Since there is this obstetric dilemma, is anything known about maternal complications/deaths at delivery or thereafter in the Maasai and Bugando women and, if so, were there differences?

Minor points

Were neonates born at Bugando Medical Centre sampled during the dry season?

Table 2: give decimals for HC in Bugando males

Some typo’s here and there.

Further point

Authors have in my view satisfactority addressed issues raised by previous reviewers

7. PLOS authors have the option to publish the peer review history of their article (what does this mean?). If published, this will include your full peer review and any attached files.

Reviewer #2: No

Reviewer #3: Yes: Cuno Uiterwaal

---

## [Author Response · Author response to Decision Letter 1]

4 Jul 2020

Guide to Revisions, PONE-D-19-25660R1

Overview: We are grateful to the reviewers for their careful reading of our paper. We agree with many points raised and have used those to improve the paper. Where issues of clarification are required we have modified the text and explained our position below. Below is a point by point description of how we have addressed the reviews.

Reviewer #2: There are significant editorial issues remaining - ranging from grammatical to spelling. I would strongly suggest that you take time to have the document propoerly edited. Spacing errors and puncttuation errors must be atteded to. Please use the same font size throughout the document (including the reference list).

Response: We agree completely with the reviewer on this and apologize for not having noticed this issue earlier. This paper has gone through many revisions and several authors have edited and re-edited the paper. This is not an excuse but it does explain why the last version got away from us the way that it did. We have thoroughly edited the paper for language and logical flow. This means that it is not really possible to highlight specific bits of text that are changed as virtually the entire paper has been intensively edited. We thank the reviewer for pointing this out.

Reviewer comment: I struggled with your comparison to the Bugando maternal-child dyads - these are non-Maasi peoples with very distinct 'tribal' variances that you have not corrected for. I am not convinced you actually can make this a reasonable comparison. Even the urban/per-urban factor was a further confusion, as the NCA residents are rural pastoralists. i remained confused as to why you did this component of the study given the minimal data collected.

Response: Yes, this is a good point. We included the Bugando sample simply because as we were conducting our study at Endulen, we were struck by the paucity of comparative data from African populations. That remains a problem. We did not intend this cohort to serve as a control sample because there are obviously many factors that distinguish these groups. Our primary comparison is to the WHO reference data. We do feel that inclusion of the Bugando data has value, if only to illustrate how strikingly the Endulen newborns deviate from the WHO norms. An ideal comparison might have been Maasai newborns from a community in which the third trimester dietary restriction was not practiced. We did try to identify such a community in Tanzania but were unable to identify one. The issue with the appropriate control remains and it is now discussed as a caveat in the discussion. We hope that this is acceptable as we believe that the findings are sufficiently striking to overcome this limitation in the research design of our study.

Reviewer comment: your consnet process also remains questionable. I would want more details about the TBAs role as my knowledge of this population (where I have done many studies) would indicate a concern with the TBAs literacy. So if you are asking them to read the consents to the participants this is highly unlikely from my experience. I would stress that you explain what was actually done.

Response: We have added text to clarify the precise role that the TBAs played in ours study design. Relying on the TBAs for the consent process was an issue that was extensively discussed with CHREB (the UCalgary) IRB at the time of study design and with NIMR. This was a significant issue in launching the study as we could not implement normal consenting processes for precisely the reasons that the reviewer identifies. We also certainly understand the reviewer’s concern as this was an issue that we discussed at length. Given the logistical and, more importantly, cultural challenges involved (which we suspect that the reviewer understands, based on the comments), there was really no other way to recruit the subjects and obtain the anthropometric data required for this project than by training and relying on the TBAs. In the end, the system that we implemented relied on verbal consents that were overseen by a local translator (Godwin Olesaningo) who is also a long-standing member of our research team for this and related work. Mr. Olesaningo trained the TBAs in conjuction with Mr. Powell, whose M.Sc. work was based on this research. Mr. Olesaningo served at the time as the community outreach coordinator for Endulen Hospital and his primary contacts for maternal health in the community were the group of TBAs recruited for this study. Paula Gonzalez was also present in the field but only for a portion of the study period. So, this creates the additional complexity that two male researchers, even if one of them is Maasai, would have virtually no direct access to the mothers recruited to this study for reasons particular to Maasai culture. The training of and reliance on the TBAs was the only way that these very sensitive data could be obtained. This means that we have records of the verbal consenting process and training of the TBAs but not for the mothers themselves. We fully realize that this is unusual and that it poses ethical questions about the validity of consent processes in non-literate, closed cultural settings such as the one in which this study occurred. CHREB approved this unusual process only with the provision that it would also be approved by NIMR. NIMR did approve our process as described and we have the documentation for all of this deliberation. While we completely understand the concern of the reviewer, we respectfully ask that they accept the deliberation and decision of the ethics boards that oversaw our work. 

 An advantage of having worked with the TBAs in this way, is that they became much more actively engaged in the work than otherwise. This is important as future interventions in this and other similar communities must leverage the care and expertise that the TBAs provide. This, of course, raises a different ethical question as to why, if the TBAs were involved via the Focus groups and training sessions in the design and conduct of this research, are they not authors on this paper. This is also an issue that we struggle with as our preference would certainly be to include them as authors. However, as the reviewer states, these women, whilst knowledgeable and intellectually capable of engaging with the work on a verbal basis, are not literate. They would not be able to meaningfully engage with the written manuscript. Further, there is content in this paper that is potentially controversial, as it relates to ongoing food insecurity within the NCA which is exacerbated by policies of the NCA itself. This would create risk for these women that is not commensurate with the small benefit of authorship on this paper.

Reviewer comment: The dietary recoall for 9 months is much more is a high expectations. The dietary intake is only one aspect of the nutrition in this gorup - for example, herbs to cause emesis(vomitting) and what quality of food is actually allocated has not been addressed in your paper. I am concerned about this possible misrepresenation.

Response: This is an excellent point. We provide caloric estimates based on the FFQ data but we agree that we should be more transparent about the limitations of these data. We have added a discussion of these issues in the Discussion session as caveats for the interpretation of these results.

Reviewer #3: Interesting and well written report on the association between third trimester maternal food intake restriction and birth sizes. The report is transparent with regard to the methodologies applied and possible alternative explanations, but there are some issues to address further.

Reviewer comment: Authors rightfully discuss social desirability as a possible source of bias with TBAS’s. However, could there also be another source of bias? TBA’s collected maternal dietary data 2-3 days postpartum, presumably having already measured birth sizes. How exactly do socially desirable answers play a role once the child was already born alive and presumably healthy albeit (too) small? A second question is if the knowledge of birth sizes could have influenced measurements through FFQ’s? Or, for that matter, have there been occasions where FFQ’s were taken before birth size measurements, where knowledge of birth sizes may have influenced the actual birth measurements? If this kind of information bias, which cannot be analytically solved, could have occurred it should be briefly discussed in the limitations section.

Response: This is a good point. We have added a discussion of this issue in lines 484-490. Specifically, we agree that it is possible that a TBA might wish to ‘protect’ a mother under their care and that this might bias the dietary data to be more consistent with an expectation. It is correct that there is ‘local knowledge’ that pregnant women ‘should’ decrease food intake and change the type of food they eat. However, it would take some level of ‘sophistication’ to alter, for example, the carbohydrate component and not some other component. As mentioned in the next section, it is also unclear if the TBA would know that the newborn is ‘small’ according to the WHO. They would really only know how the newborn compared to other newborns in the area. So, we do not think there is a bias in that regard. 

Reviewer comment: Since there is this obstetric dilemma, is anything known about maternal complications/deaths at delivery or thereafter in the Maasai and Bugando women and, if so, were there differences?

Response: This is an excellent point about which we have also wondered. Unfortunately, there are no data from Endulen on rates of obstetric complication so it is completely unknown whether the dietary reduction strategy actually accomplishes its purported goal. The vast majority of Maasai births are at home and birth complications are not systematically reported in any way, so these data do not exist, unfortunately.

Minor points

Reviewer comment: Were neonates born at Bugando Medical Centre sampled during the dry season?

Response: No, these data were collected in December of 2014 and January of 2015. There would not be a similar seasonal fluctuation of food availability in Mwanza as in Endulen, however. See comments about the relevance of this comparison above.

Reviewer comment: Table 2: give decimals for HC in Bugando males

Response: This has been fixed.

Reviewer comment: Some typo’s here and there.

Response: We have thoroughly edited the paper for grammatical issues and logical flow.

e

---

## [Decision Letter · Decision Letter 2]

3 Aug 2020

Lack of head sparing following third-trimester caloric restriction among Tanzanian Maasai

PONE-D-19-25660R2

Dear Dr. Gonzalez,

We’re pleased to inform you that your manuscript has been judged scientifically suitable for publication and will be formally accepted for publication once it meets all outstanding technical requirements.

Kind regards,

Jai K Das

Academic Editor

PLOS ONE

Additional Editor Comments (optional):

Reviewers' comments:

Reviewer's Responses to Questions

**Comments to the Author**

1. If the authors have adequately addressed your comments raised in a previous round of review and you feel that this manuscript is now acceptable for publication, you may indicate that here to bypass the “Comments to the Author” section, enter your conflict of interest statement in the “Confidential to Editor” section, and submit your "Accept" recommendation.

Reviewer #2: All comments have been addressed

Reviewer #3: All comments have been addressed

2. Is the manuscript technically sound, and do the data support the conclusions?

Reviewer #2: Yes

Reviewer #3: (No Response)

3. Has the statistical analysis been performed appropriately and rigorously? 

Reviewer #2: Yes

Reviewer #3: (No Response)

4. Have the authors made all data underlying the findings in their manuscript fully available?

Reviewer #2: Yes

Reviewer #3: (No Response)

5. Is the manuscript presented in an intelligible fashion and written in standard English?

Reviewer #2: Yes

Reviewer #3: (No Response)

6. Review Comments to the Author

Reviewer #2: Thank you for your considered corrections/edits in response to my previous comments and suggestions. I think your paper is much stronger and reflects more clearly the research processes and rationales. There remain a few very minor edits and I would encourage you to take a last read through to ensure you make these very minor corrections.

Reviewer #3: (No Response)

7. PLOS authors have the option to publish the peer review history of their article (what does this mean?). If published, this will include your full peer review and any attached files.

Reviewer #2: **Yes: **Pammla Petrucka

Reviewer #3: **Yes: **C.S.P.M. Uiterwaal

---

## [Editor Report · Acceptance letter]

6 Aug 2020

PONE-D-19-25660R2 

Lack of head sparing following third-trimester caloric restriction among Tanzanian Maasai 

Dear Dr. Gonzalez:

I'm pleased to inform you that your manuscript has been deemed suitable for publication in PLOS ONE. Congratulations! Your manuscript is now with our production department. 

Kind regards, 

on behalf of

Dr. Jai K Das 

Academic Editor

PLOS ONE